# Tobacco Tax Increases: A Discourse Analysis of the French Print and Web News Media from 2000 to 2020

**DOI:** 10.3390/ijerph192215152

**Published:** 2022-11-17

**Authors:** Diane Geindreau, Morgane Guillou-Landréat, Karine Gallopel-Morvan

**Affiliations:** 1CNRS, Inserm, Arènes—UMR 6051, RSMS—U 1309, EHESP, Université de Rennes, 35000 Rennes, France; 2Addictive Disorders Department, EA SPURBO, Université de Bretagne Occidentale, 29238 Brest, France

**Keywords:** tobacco tax increases, discourse analysis, French general press, arguments, tobacconists

## Abstract

Lobbying led by the tobacco industry and tobacconists is a barrier to reducing smoking prevalence in France. Here, we analyze the discourse of the tobacco industry and other key actors (public health agencies, politicians, etc.) in the French general-audience news media from 2000 to 2020 around tobacco tax increases, which is one of the most effective tobacco control measures, especially amongst youth. We queried Europresse (a European news media and specialized press database) using the keywords “increase”, “price or taxes”, and “tobacco or cigarettes”, and found 5409 topic-relevant articles, from which we extracted 8015 arguments for or against the measure. In total,64.3% were against the measure (mostly on grounds of “ufueling the black market”), 32.1% were for the measure (mostly claiming it is “effective at reducing smoking prevalence”), and 1.8% proposed alternative measures. Tobacconists, the primary source of media content on the topic, led a discourse that was strongly opposed to tax increases. Public health agencies, which attracted only half as much media attention, were strongly supportive of the measure. Analysis of discourses relayed in the French general-audience press revealed overwhelming opposition to tobacco tax increases, and this discourse was widely advanced by tobacconists. The results were congruent with international literature that had highlighted a similar set of arguments to those found in the French general press that were broadcasted by the tobacco industry and its allies (tobacconists in France) in an effort to block this evidence-based public health measure.

## 1. Introduction

Tobacco kills more than 8 million people worldwide each year [1]. In a concerted effort to control this epidemic, the WHO devised the Framework Convention on Tobacco Control (FCTC) [2], which consists of evidence-based recommendations for measures that signatory countries are expected to adopt to reduce smoking prevalence. Article 6 of the FCTC probes “Price and tax measures” to reduce demand for tobacco and urges signatories to “recognize that price and tax measures are an effective and important means of reducing tobacco consumption by various segments of the population, in particular young persons” [2].

The literature on article 6 of the FCTC finds that tax policy is a cost-effective measure of tobacco control [3,4]. It has been shown that maintaining low prices is an incentive for tobacco use and an argument advanced for marketing purposes, especially in contexts where advertising is banned [5]. Conversely, repeated tax increases are a measure that makes tobacco products less affordable and desirable, increases attempts to quit, and helps prevent young people starting smoking, which will eventually decrease global demand for tobacco products [6,7]. Price-elasticity analyses in high-income countries show that a 10% increase in the retail price of tobacco will lead to a 4% decrease in sales, with youth and young adults being the most responsive segment [7]. However, research has found that any increase below this 10% threshold is ineffective at reducing demand for tobacco. Finally, tax increases also provide revenues that governments can earmark for further tobacco control or public health measures.

As tax policy is effective in reducing demand for tobacco, it hits the profits of the tobacco industry (TI) [3]. To preserve its profits and defend its interests, the TI practices lobbying in an effort to influence tax level or structure [3,6,8]. In a review of 36 studies published between 1985 and 2010, Smith et al. classified these practices into arguments and tactics [3]. First, the TI uses a wide range of arguments to keep taxes down at the lowest level. The most frequent arguments advanced are: (1) price increases will lead to illicit trade [3,7,8], (2) tobacco taxes are regressive (tobacco taxes eat up a higher percentage of the income of poorer consumers, so raising taxes is unfair to them) [3,7], (3) governments will lose revenue [3,7,8], and (4) the public would not support the policy [3]. Second, as it cannot leverage its own compromised reputation, the TI seeks to recruit third-party actors such as front groups or more credible allies (labor unions, tobacco sellers, scientists, or ideologically-motivated groups) to defend its interests, typically through press releases and commissioned studies [3,8,9]. They also sow confusion in the population around the purposes and effectiveness of tobacco tax increases by appealing to wider concerns around taxes or political distrust [3]. Finally, the TI positions itself as an expert ready to bring decision makers solutions for supposedly fairer and more cost-effective excise structures.

There is fairly solid literature on arguments and strategies used by the TI to counter the adoption of tobacco tax increases [3,10], but there has been little effort to analyze the arguments and strategies spread more generally by front groups, health representatives, and other stakeholders. The aim of our research is to identify the volume of TI discourse on this topic in tandem with discourses by other stakeholders such as front groups and politicians, and how supporters of tobacco tax measures manage to counter the TI’s arguments.

The literature on lobbying by the TI (and other actors) against/for price increases remains mostly contextualized within Anglosphere countries [3,6,10]. However, it has been shown that the TI adapts its strategy to the target market [3], which raises the issue of generalized arguments and tactics revealed by Smith et al. and Ulucanlar et al.

The research reported here is conducted in Europe, where the “Tobacco Tax” Directive 2011/64/EU (TTD) defines a minimum excise duty rates (between 7.5% and 76.5% of the total tax burden) and an ad valorem tariff expressed as a percentage of the maximum pack price [11]. The TTD was recently evaluated and found to perform ineffectively on deterring tobacco use, and so the European Commission is currently considering a more ambitious harmonized taxation policy [12]. As the TTD is not fully harmonized, some Member States have used specific tobacco policy taxation embedded in the Directive.

This research reports the case of France, which has a national tobacco taxation policy and its own specific tobacco-related context, as described below.

First, smoking prevalence is high, with 25.5% of 18–75 year-olds smoking daily in 2020 [13], which is higher than the average in very high Human Development Index countries [14]. This high prevalence is, among other reasons, a result of lobbying by the TI and its commercial partners in France, the tobacconists [15].

Second, despite ambitious national plans since 2014 (the National Program to Reduce Smoking in 2014 and the National Tobacco Control Strategy in 2018 [16,17]) that included measures such as plain packaging and social marketing campaigns (“Mois sans tabac”), the application of FCTC article 6 on tobacco tax in France remains erratic. Since 2000, the French tobacco taxation strategy can be divided into five key periods [18] (Table 1): from 2003 to 2004 (period 1), the National Anti-Cancer Program [19] increased tobacco taxes by 40% in total. From 2004 to 2007 (period 2), the French government adopted a tobacco tax moratorium [20], under pressure from tobacconists who were firmly opposed to further tax rises. From 2007 to 2014 (period 3), prices increased continuously but too slowly—5% to 6% increases, i.e., below the effective 10% threshold—to meaningfully reduce demand for tobacco [4]. From 2014 to 2018 (period 4), prices remained stable, at around EUR 7 for a pack of cigarettes). Finally, from 2018 to 2022 (period 5), the second National Tobacco Control Strategy roadmap set a price of EUR 10 for a pack of cigarettes, which was reached in 2020 through successive increases [17]. Unlike period 3, the roadmap to reach the price of EUR 10 per pack included increases above the 10% threshold required to effectively reduce demand for tobacco.

Tax policy was in line with article 6 of the FCTC during periods 1 and 5 but not in periods 2 to 4 inbetween.

France poses a specific market context, with a monopoly on tobacco sales (similar to Spain, Belgium, and Hungary [21]) owned by a network of 24,000 tobacconists under contract with the State. These actors are very close to the TI, to the point that they are suspected to act as a TI front group [22].

The French government’s accountability office officially reported that France’s erratic policy strategy on tobacco tax is explained by strong lobbying deployed by the TI (La Cour des Comptes [15]). However, no research has yet explored the arguments used by the TI and its allies (including tobacconists), or by other stakeholders in this high-smoking-prevalence country.

To address this gap, here we set out to identify and analyze the arguments disseminated through press coverage around tobacco tax increases in France over the 21-year period characterized by erratic tobacco tax policies (Table 1). Media analysis is relevant here as a way to study debate around health issues and the arguments used by businesses. First, it has been shown that the TI used news media to disarm tobacco control advocacy messaging, either directly or by using front groups [10,23]. Second, the press has the power to influence public debate on policy adoption by raising awareness on a specific issue and by exposing readership to opinions and arguments from various parties [24]. Newspapers also play an agenda-setting role as they lend relevancy to issues through editorial decisions on which health issues to cover and which ones to not [25,26].

The questions asked in our research are: (i) What is the scale of coverage on tobacco tax increases in the French general press during the five key periods (Table 1)? (ii) What kind of arguments get disseminated, and do the arguments change over the five key periods? (iii) Who spreads these arguments (tobacco industry, tobacconists, NGOs, political or other actors) and what is the valence of the arguments they use? and (iv) Are the TI and front-group arguments identified in France similar to or different from the arguments identified by Smith et al. in the international literature?

## 2. Materials and Methods

### 2.1. Data Collection

Content analysis was conducted on articles about tobacco tax increase published in the French general press between 1 January 2000 and 31 December 2020. Articles were found using Europresse (https://nouveau.europresse.com/Search/Reading, accessed on 17 March 2021), a European press archives database that provides access to articles from the general-audience and specialized press (paper copy and web platforms).

The search string “TIT_HEAD=tobacco|cigarettes& TEXT= increase+$3(price|taxes+) was run and produced 7974 hits including both newspaper and web platforms. The articles were all read and then selected or not according to the following inclusion criteria: articles from the French press only (no other countries), local to international coverage, from 2000 to 2020, and mentioning tobacco tax increase or price increase through taxation in France, either in the headlines or in the body copy. Exclusion criteria were articles that referred to tobacco taxation policies in other countries or other tobacco control measures. If duplicates were found but from different sources (paper-print and web platform), both were included as they were considered as targeting different readers. Any doubts around whether articles were relevant were resolved by double-checking with the other authors of the paper. A final total of 5409 articles were included in the study (Figure 1).

The 5409 articles identified were analyzed to evaluate the volume of press coverage on that topic. In total, 1908 articles did not contain any argument for or against a tobacco tax increase; they were factual and neutral and therefore excluded from the second part of the study, i.e., qualitative analysis. In total, 3501 articles that vectored arguments were included in the qualitative analysis, and 8015 arguments were coded based on our codebook classification; 5114 of these arguments were delivered by an identified speaker, and 2901 were used by the journalist with no reference to any identified speaker (Figure 1).

An Excel spreadsheet was used to compile all the meta-characteristics of each article, including source, author, title, nature of the article, and type of newspaper (local, regional, national, international). All articles were publicly available and did not therefore require ethical permissions.

### 2.2. Analysis

A quantitative analysis using Microsoft Excel was conducted on the 5409 articles included in the study. The frequency of published articles was calculated per year and grouped into the five key periods of tobacco taxation policy identified in France (Table 1).

After the first round of read-through, we conducted a quantitative analysis and a qualitative analysis. The qualitative analysis consisted of a thematic content analysis on the articles carrying arguments. Using an inductive approach, which consists of analyzing the data without trying to make it fit to “pre-existing concepts or ideas from theory” [27], we produced a codebook layered into a stance on the topic, main argument, and sub-arguments (Appendix A). All articles were scanned through the lens of the codebook to count occurrences of arguments and look for potential patterns in the use of these arguments.

The speakers on tobacco taxation were also identified when information was available, and then classified by organizational category, i.e., TI, tobacconists, politicians, public health NGOs, or medical (Appendix A). The frequency and valence of their arguments were calculated in Microsoft Excel on a scale from −1 for “strongly opposed” to +1 for “strongly in favour”. Any doubts around the coding process were resolved by double-checking with the other authors of the paper.

## 3. Results

### 3.1. Volume of Articles

The 5409 articles centered on tobacco as a topic included in the quantitative analysis showed three peaks in publication rate (2003, 2013 and 2017; Figure 2).

The peaks in 2003 (period 1) and 2017 (period 5) corresponded to substantial tax increases adopted in France. The peak in 2013 (period 3) arrived at the end of period, 3 which was characterized by small but continuous incremental tax increases. In 2013, the pack price for cigarettes reached EUR 7 and the market lost 2.5% in value. In this context, detailed analysis of the arguments extracted for 2013 showed that anti-tax voices asserted that the increases were too frequent and had negative economic impacts, whereas pro-tax voices regretted that the increases were too timid, i.e., not high enough to have a real impact on attempts to quit and on smoking prevalence.

Conversely, the years where the volume of articles published was lowest correspond to periods when there was a lull in tax increases (period 2) or years following tax increase measures adopted (period 5).

### 3.2. Qualitative Analysis

#### 3.2.1. Identified Arguments

The leading category (64.3%) consisted of arguments against tobacco tax increases, subdivided into four main arguments: (A) tax increase is ineffective; (B) tax increase causes social and economic damage; (C) tax increase disrupts the market; and (D) tax increase has negative impacts for smokers in their day-to-day life (Table 2).

The main argument “A” was divided into six sub-arguments/reasons explaining why increasing tobacco tax is ineffective (from most to least frequent): “raising tax will have no effect on tobacco use”, “smokers don’t quit but switch to cheaper products”, “lower sales do not mean that people have quit smoking”, “smokers anticipate increases by buying in more”, “smokers do not quit smoking but switch to vaping”, and “smuggling and cross-border markets make tobacco more accessible to youth”.

The main argument “B” was divided into 11 reasons arguing that the measure would harm individuals and the State (from most to least frequent): “tax increases will hurt tobacconists”, “tax increases will provoke security problems for tobacconists and delivery services”, “tax increases will hurt border tobacconists”, “there is a loss of revenue for the State”, “tax increases will hurt manufacturers”, “it is the State’s responsibility to support tobacconists in adapting to tax-related changes”, “Tax increases risk prompting anti-government votes from tobacconists”, “tax will hurt producers”, “there are adverse effects (without giving further details)”, “tax increases will hurt Corsican tobacconists”, and “tobacco tax is not a reliable source of revenue for funding important social or public health programs”.

The main argument “C” was divided into six reasons that demonstrated how a tax increase would hit the economy (from most to least frequent): “tax increases fuels the dark market, smuggling, illegal trade”, “it increases legal purchases in border countries”, “tax increase is not a public health measure but a way to pay off State debt”, “it increases the price differences between France and neighboring border countries”, “it increases purchases made online”, and “lockdown in 2020 led to an increase in sales at tobacconists, which means that clients who were purchasing abroad went back to local retailers (proof that they had not quit but simply adapted their purchasing practices).

The main argument “D” was divided into four reasons advanced to claim the measure was detrimental to smokers, marginals, and the poorer (from most to least frequent): “taxes are regressive and unfair to the poorer”, “taxes are unfair and punitive towards smokers”, “taxes are one of the triggers for the “gilets jaunes” social movement”, and “taxes are intrusive and constrain smokers’ freedoms”.

The second category counted three main arguments supportive of the measure (32.1%): “E” tax increase is an effective measure to reduce smoking, “F” “tax increase benefits the different parties”, and “G” “some information on negative effects of tax increase is inaccurate or exaggerated” (Table 3).

The main argument “E” was divided into six reasons asserting why the measure is effective under certain conditions (from most to least frequent): “tax increase is an effective measure to reduce smoking or increase quit attempts”, “it is effective if increases are >10%, continuous, regular, and coupled with other tobacco control measures”, “when too weak (<10%), the measure is not effective in reducing smoking but just a way to enrich the tobacco industry—i.e., it is not a public health measure” (this argument defends the adoption and efficiency of heavy taxes), “tax increase is effective if prices of different tobacco products are standardized”, “tax increase does not have the same efficiency across all the population, “tax increases are more or less dissuasive”, and “it is the tobacco industry agreement on prices that mitigates the efficiency of the measure”.

The main argument “F” was divided into three pro-measure reasons, arguing that it benefits the different parties, which runs against conventional wisdom that it would add to the ongoing economic crisis (from most to least frequent): ”tax increases add to tobacconists’ revenue”, “tax increases add to the tobacco industry’s revenue” (the profits continue to compensate for the drop in sales), and “tax increases raise government funds for public health programs”.

The main argument “G” was divided into three reasons discussing the fact that some of the arguments for the measure are not evidence-based (from most to least frequent): “there is no proof of a link between the measure and smuggling or the dark market”, “there is no proof of a link between the measure and increased risk for the tobacconists”, and “the general population is supportive of the measure”.

The third category of arguments on the tobacco tax increase counts two proposals for alternatives or prerequisites to the measure: “H” “the taxation strategy alone is not effective to reduce smoking—it needs to work on the back of other measures” and “I” “If kept moderate, a tax increase will be beneficial to all” (defends an increase around 5%) (Table 4).

The main argument “H” countedfour reasons arguing that the measure would be useless without other measures taken beforehand, such as (from most to least frequent) harmonized prices across the EU, measures to combat smuggling, prevention measures addressing youth, and advocating lower-risk products.

Finally, the main argument “I” proposed to limit the increase (to around 5%) for the reason that it would be more beneficial to all (State, tobacconists, and the industry) as it would increase profits, provide funds for the State, and be more acceptable (or less unfair) for smokers.

Overall, the category of arguments against the measure was strongly dominant over the 21 years period of the study, except in the years 2000, 2001, 2015, and 2016 when the category of arguments for the measure was slightly more frequent. The leading sub-argument was that the measure would drive the growth of cigarette smuggling and the black market. Appendix A) show that the use of this sub-argument remained constant over the years, with an intensification in 2003 (*n* = 257), 2013 (*n* = 146), and 2017 (*n* = 171), which were years marked by legislative stakes. The second most frequent sub-argument (or reason) was that a tax increase is effective in reducing smoking. This sub-argument was intensively used in 2003 (*n* = 182) and 2004 (*n* = 112), and then practically disappeared before re-merging in 2013 (*n* = 103) and around 2017 (*n* = 114). The third most frequent sub-argument was that the measure would hit tobacconists hard, and was used constantly over the 21 years of the study, with a major peak in 2003 (*n* = 219) and another, less intense peak in 2017 (*n* = 91).

#### 3.2.2. Number and Valence of Arguments per Speakers

We led an analysis of the speakers that had voiced the identified arguments. A total of 2448 articles made it possible to identify specific speakers through 5114 citations of various people and/or organizations (Figure 1).

We identified 15 broad sets of speakers or organizations., ranging from lay commentators (anonymous) to different kinds of specialists in the fields of medicine, public health, economics, customs or law enforcement, and including all the actors involved in tobacco production and sale, from producers to industry and on to retailers (tobacconists), and the politicians expected to enact the measure.

Citations for these speakers were counted and analyzed (Figure 3). Their discourses in the French press were scored on a scale from −1 for “strongly opposed to tobacco taxation” to +1 for “strongly in favour of tobacco taxation”. The tobacconists were the most vocal category, with 2368 citations. They were strongly against the measure with a valence of −0.92, which made them the strongest opponents to the measure. In second position were the politicians (government and regulatory bodies), with 899 citations. Their discourse leaned in favour of the measure, with a valence of +0.30. In third position were the bodies representing health professionals and public health agencies, with 853 citations and a discourse strongly in favour of the measure with a valence of +0.87, and they were the strongest supporters of the measure. In fourth position was the TI with 460 citations and a discourse strongly opposed to the measure with a valence of −0.80.

Appendix A) shows that the tobacconists were the leading voice on tobacco tax increases for all 21 years of the study except 2001, 2010, and 2016. These years correspond to period 1, period 3, and period 4, which are periods in which there were little or no cigarette pack price increases. Tobacconists, politicians, public health representatives, and the TI all intensified their volume of comment in the news media in 2003, 2013, and 2017, which were periods marked by intense debate around tobacco control (tobacco taxation for 2003 and 2017 and plain packaging for 2013).

## 4. Discussion

A tobacco tax increase is one of the most effective measures for preventing starting smoking, increasing quit attempts, and decreasing smoking rates [3]. Research has shown that the TI has historically responded to tax increases by using a discourse-based strategy to block the adoption of the measure [3,10], especially through using media attention [9,28].

Here, we sought to analyze the discourse on tobacco tax increases vectored through the French general press over a 21-year period. Arguments around the taxation measure (for, against, and alternatives) were extracted, counted, and situated in terms of key periods for tobacco control in France and the speakers voicing the discourse. Arguments “against” the measure were overwhelmingly dominant. Globally, all kinds of arguments increased in volume in the years 2003, 2013, and 2017, which were key periods surrounding the debate on tobacco control, but the balance remained essentially unchanged with arguments “against” continuing to predominate.

Similarly to Smith et al. and Ulucanlar et al., the main arguments against increasing tobacco tax in France were a claimed increase in smuggling and illegal trade and the risk of loss of revenue and jobs. Research from Nepal and Malawi [29,30] assessing the idea of a globalized discursive strategy deployed by the TI showed similar findings [30]. Arguments asserting that the tax was regressive were less significant in the French context (1.5%) than in the Anglosphere context described by Smith et al. [3]. Smith et al. also identified an entire category of arguments based on the concept of “earmarked taxes” that were not found in the French general press, showing how discourse on tobacco tax increases can be shaped by legal, fiscal, and cultural landscape. Indeed, while there were similarities between arguments against the taxes in the French and Anglosphere contexts, our study revealed that global arguments were to a certain extent re-shaped and adapted to local context. In France, the argument around negative impact on workers and local economy took shape around the figure of the tobacconists, who are argued to be endangered by adoption of the taxation measure though illegal trade and unfair competition due to geographical position in the Schengen area, further complicated by security problems. Otañez et al. also found that the global economic argument was adapted to the context in Malawi by underlining the importance of tobacco farming for the country’s autonomy and for reduction of poverty alleviation [29].

Beyond the arguments around tobacco tax increases, our research also identified the key emitters of the discourses and their stance on the taxation measure in the news media. Tobacconists, who have a monopoly on tobacco sales in France, were the most productive voice for arguments on tobacco tax increases, and they were also the strongest opponents of the measures. Conversely, the TI was five times less active in its communication on tax measures as compared to tobacconists. However, the two constituencies shared a common narrative and, on several occasions, they spoke as one voice. In their case study on industry front groups, Apollonio et al. showed that reputational compromise prompted the TI to use front groups as a way to advocate in favour of their interests. They demonstrated that seemingly independent organizations were developed to combat tobacco control regulations and draw popular support, especially by acting as representatives of the “grass-roots level” [9]. Our study showed that the argument of a certain proximity with the people was one of the most salient parts of the discourse spread by tobacconists in France through news media. Ulucanlar et al. also showed that these front organizations strongly supported the TI through lobbying or media agenda-setting, thanks to the media training they were provided with [10], which is further evidence that the news media is central to TI strategies to mitigate the influence of advocates for tobacco control. In France, the tobacconists community was not constructed by the TI and there is no direct proof that they act as a front group for them. However, as mentioned, they do share a very similar discourse on tobacco tax increase in the French general press. Szilágyi et al. demonstrated that beyond financing seemingly independent groups, the TI also forged strategic connections with natural allies, who are parties willing to defend the same positions because they share common interests [28]—in this case, tobacconists and the TI. These elements prompt a move to dig deeper into the relations between these two organizations.

A frame is a way to select and “make salient certain aspects of perceived reality in such a way as to promote a particular problem definition” [31]. Tobacconists benefit from a more positive image in France than the TI, supported by a strong communication strategy [32]. They frame themselves as close to the people and their concerns, especially in increasingly under-served rural areas where they maintain public services, social support, and conviviality [22]. In response to increasingly tough tobacco control policies, they developed a frame, destinated to media, based on victimization and catastrophism in an attempt to defend their interests [22]. Taxation is shaped as targeted persecution, hurting the tobacconist’s network and further driving the economic crisis. Ulucanlar et al. describe such practices, the creation of an alternative reality, as a discourse-based strategy used by the TI with the aim to shift the debate away from smoking harm and tobacco taxation towards core values (proximity, rurality, tradition, employment) that are being jeopardized [10,33].

As tobacconists are the predominant voice in the news media, the economic discourse that they tend to hold can be defined as a default frame, with a strong narrative to influence the public’s perception around the problem of taxation and the required solution [31]. It may influence individuals’ opinions, attitudes, behaviors, and norms. Media framing is also “the basis by which public policy decisions are made” [33]. Two studies in the US, which currently has a fairly low prevalence of smokers (12.5% in 2020, https://www.cdc.gov/tobacco/data_statistics/fact_sheets/adult_data/cig_smoking/index.htm, accessed on 24 June 2022), showed that media framing in newspapers on tobacco tax was mostly pro-increase [34,35]. However, Harris et al. also showed that the amount of articles against the measure dramatically increased a month before a crucial vote, and ultimately led to a rejection of the tax increase [35]. The volume and content of the articles published are an incentive for politicians to take a vocal stance on a public health problem [33]. In France, media attention is mainly focused on the supposed threat posed by the measure. A direct impact on voting intentions remains unproved, but the repeated framing of a problem contributes to the impression of public acceptability of a measure, which could indirectly influence political decisionmakers [36].

Overall, our evidence showed that the French general press vectored a discourse that was mostly against tobacco tax increases. These results are highly valuable as they make salient the great unbalance that exists in the debate. This imbalance penalizes public health advocates, which can diminish the acceptability of their discourse and ultimately potentially postpone the adoption of ambitious new measures, especially in a context of price stagnation since reaching EUR 10 per cigarette pack. This conclusion should prompt public health advocates to reinforce their media advocacy strategy by: managing the volume of its communications in the news media; preparing ahead for key periods when opponents to pro-health measures can be expected to step up their own communications (timing); elaborating counter-arguments addressing the “default frame” that claims the measure would benefit illegal trade and border countries to the detriment of the tobacconists, who are the last small businesses still open in under-served communities.

This study carries several limitations. The selection process may have missed some articles, but the size of the sample suggests that more articles would not have substantially changed the results. The coding process is intrinsically related to individual perception and interpretation. Another coder could have formulated the arguments in a different manner, but when doubts emerged, discussions were held with the other authors to ensure reliability of the findings. Furthermore, this study remains limited to discourses relayed in the French general press. Triangulation with other kinds of discourses, from different types of documents such as parliamentary papers or trade press (to a tobacconist audience) would be of interest to identify potential new arguments, analyze the congruence of the findings, and evaluate the influence of the arguments in the legislative process. Another limitation is that our research is France-centered. France is part of the wider the European Union where the TTD is currently being revised. Some researchers and public health figures propose using the TTD revision to converge prices and adopt an affordability-based minimum tax [37]. In this context, it would be instructive to monitor the main European media channels to capture the arguments developed by the tobacco industry to counter this proposal.

## 5. Conclusions

Previous research on tobacco tax increases has shown that the measure is a useful instrument for reducing demand for tobacco products [3,4]. The tobacco industry and its allies have attempted to curb the measure by releasing arguments such as: it will increase the black market and smuggling, it is regressive and unfair to the poorest, and it will provoke a loss of jobs and revenue [3,10]. The tobacco industry intensively exploits news media as a channel to spread these arguments [10]. However, these previous studies were mainly contextualized within Anglosphere countries and few of them have analyzed other types of arguments (pro-tax) and other voices speaking about it (politicians, public health representatives, allies, academics etc.). The aim of this paper was to examine the different kinds of arguments used around tobacco taxation increases spread by different speakers (opponents and supporters) through the French general press. We showed that the pace and volume of publications on the topic were in step with the legislative schedule on tobacco tax policy. We then demonstrated that a whole range of arguments around the measure was spread in the news media, with arguments standing against the adoption of tax increases being overwhelmingly dominant in frequency but also in variety. We identified various different voices on the measure in the news media, with tobacconists being the most productive of them. This organizational category proved “strongly opposed to tobacco tax” increases by speaking out frequently and using a range of counter-arguments that showed overlapping content with the arguments used by the tobacco industry (as identified by the literature) [3,10]. The form of their discourse also showed similarities with the tobacco industry ‘s strategies, especially the use of ideological arguments focused on touting imminent disaster if the measure were passed [10,38], with heavy consequences that would hit their community, which they self-describe as close to people’s needs and considerations. Our research offers several contributions to the existing literature on lobbying and advocacy on tobacco taxation, and we formulate several recommendations. Media is a key tool for tobacco control advocacy, both to inform legislators and the wider public and to support the adoption of policies [39,40]. Our research reveals that French NGOs should develop more proactive and professionalized press relations: first by anticipating the coming or ongoing key legislative periods by stepping up their efforts to reach editors and journalists, second by increasing the volume of their communications (on a regular basis and even more during key periods), and finally by developing a framing that is strong enough to counteract the frame that tobacconists have successfully implemented in France and which has become a popular default frame.

## Figures and Tables

**Figure 1 ijerph-19-15152-f001:**
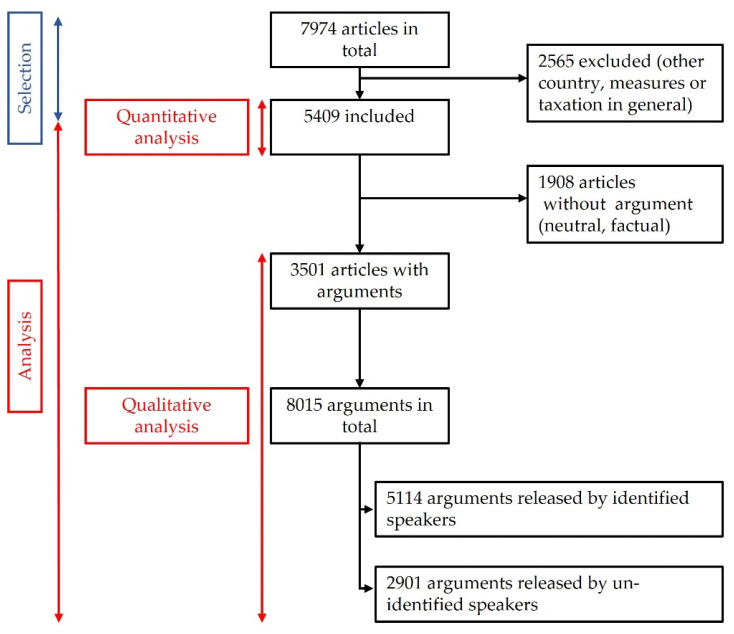
Flowchart for the article selection and argument analysis process.

**Figure 2 ijerph-19-15152-f002:**
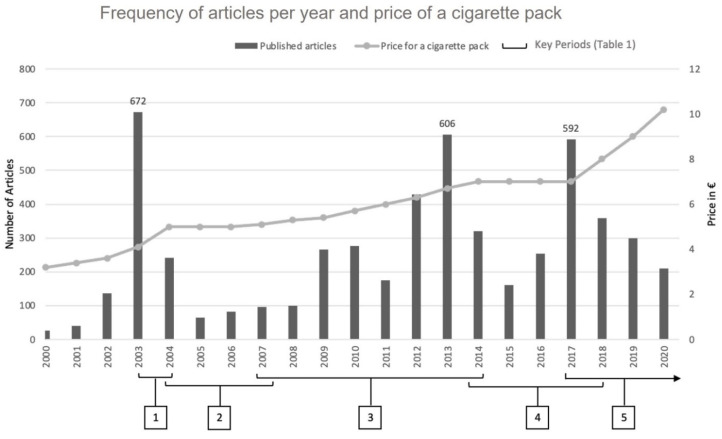
Number of articles per years with regard to tobacco price increases and with the 5 key periods of taxation policies in France between 2000 and 2020 (*n* = 5409). Key Periods source: Table 1.

**Figure 3 ijerph-19-15152-f003:**
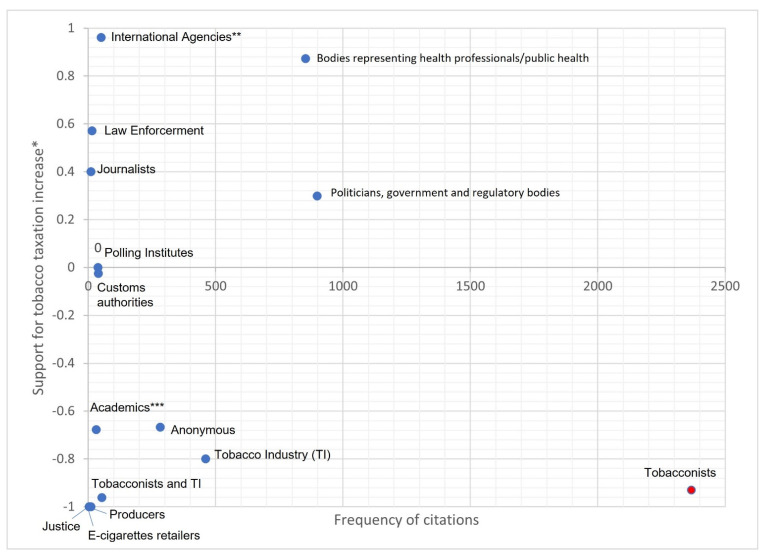
Frequency of citations per speaker-constituency and valence of their discourse. * Scored on the scale from −1 “strongly opposed to tobacco tax increase” to +1 “strongly in favour of tobacco tax increase”. ** “International” Agencies refer to the WHO and the World Bank. *** “Academics” stands for researchers working in other fields outside medicine or public health (economics, history etc.).

**Table 1 ijerph-19-15152-t001:** Key periods of tobacco tax increase in France between 2000 and 2020.

Period	Date Range	Context	Increase (EUR)	Increase (%)	Final Price for the Biggest-Selling Cigarette Packs
(1)	2003–2004	First National Anti-Cancer Program	+EUR 0.48 +EUR 0.92	+13.3% +22.5%	EUR 4.08 EUR 5.00
(2)	2004–2007	Tobacco tax moratorium	Stable		EUR 5.00
(3)	2007–2014	Slow but continuous price increases	+EUR 0.13 +EUR 0.17 +EUR 0.05 +EUR 0.30 +EUR 0.33 +EUR 0.32 +EUR 0.40 +EUR 0.30	+2.6% +3.3% +0.9% +5.6% +5.6% +5.3% +6.3% +4.5%	EUR 5.13 EUR 5.30 EUR 5.35 EUR 5.65 EUR 5.98 EUR 6.30 EUR 6.70 EUR 7.00
(4)	2014–2018	Prices reached a 7 EUR plateau	Stable		EUR 7.05
(5)	2017–2022	National tobacco control s (2018–2022): sets a price of 10 EUR for a cigarette pack	+EUR 0.83 +EUR 0.90 +EUR 1.17	+11.8% +11.4% +13.3%	EUR 7.88 EUR 8.78 EUR 9.95

**Table 2 ijerph-19-15152-t002:** Arguments against tobacco tax increase.

Category of Arguments (*n* = 8015 Arguments Spread across 3501 Articles)	Occurrences, *n* (%)
Against	5155 (64.3)
(A) A tax increase is ineffective	1029 (12.8)
Raising tax will have no effect on tobacco use	661 (8.2)
Lower sales does not mean that people have quit smoking	128 (1.6)
Smokers do not quit smoking but switch to vaping	40 (0.5)
Smokers do not quit smoking but switch to cheaper products	137 (1.7)
Smokers anticipate price increases by buying in more (before the increase comes in)	47 (0.6)
Smuggling and cross-border markets make tobacco more accessible to youth	16 (0.2)
(B) A tax increase causes social and economic damage	1727 (21.5)
There is a loss of revenue for the State	158 (2.0)
There are adverse effects (without giving further details)	11 (0.1)
Tax increases will hurt tobacconists	898 (11.2)
Tax increases will hurt border tobacconists	224 (2.8)
Tax increases will hurt Corsican tobacconists	6 (0.1)
Tax increases will hurt producers	17 (0.2)
Tax increases will hurt industry	73 (0.9)
It is the State’s responsibility to support tobacconists in adapting to tax-related changes	50 (0.6)
Tax increases will provoke security problems for tobacconists and delivery services	242 (3.0)
Tax increases risk prompting anti-government votes from tobacconists	45 (0.6)
Tobacco tax is not a reliable source of revenue for funding important social or public health programs	3 (0.03)
(C) A tax increase disrupts the market	2258 (28.2)
A tax increase fuels the dark market, smuggling, illegal trade	1340 (16.7)
It increases the price differences between France and neighboring border countries	94 (1.2)
It increases legal purchases in border countries	575 (7.2)
It increases purchases made online	92 (1.1)
Lockdown in 2020 led to an increase in sales at tobacconists— clients who were purchasing abroad went back to local retailers (they had not quit)	43 (0.5)
A tax increase is not a public health measure but simply a way to pay off State debt	114 (1.4)
(D) A tax increase has negative impacts for smokers in their day-to-day life	141 (1.7)
Taxes are regressive and unfair to the poorer	121 (1.5)
Taxes are unfair and punitive toward smokers	17 (0.2)
Taxes are intrusive and constrain smokers’ freedoms	1 (0.01)
Taxes are one of the triggers for the “gilets jaunes” social movement	2 (0.02)

**Table 3 ijerph-19-15152-t003:** Arguments in favour of tobacco tax increase.

Category of Arguments (m = 8015 Arguments Spread across 3501 Articles)	Occurrences, *n* (%)
In favour	2576 (32.1)
(E) A tax increase is an effective method to reduce smoking	2233 (27.9)
A tax increase is an effective measure to reduce smoking/to increase quit attempts	1041 (13.0)
It is effective if increases are >10%, continuous, regular, and coupled with other tobacco control measures	763 (9.5)
It is effective if prices of different tobacco products are standardized	119 (1.5)
The different tobacco brands are suspected of aligning their prices to mitigate the impact of tax increases	11 (0.1)
When too weak (<10%), the measure is ineffective for reducing smoking but just a way to enrich the tobacco industry—i.e., it is not a public health measure (the measure has to be ambitious enough)	261 (3.3)
A tax increase does not have the same efficiency across all the population. Tax increases are more or less dissuasive.	38 (0.5)
(F) A tax increase benefits different parties	254 (3.2)
Tax increases add to tobacconists’ revenue	109 (1.4)
Tax increases add to the tobacco industry’s revenue	86 (1.1)
Tax increases will raise general State revenue (dedicated to public health programs)	59 (0.7)
(G) Some information on negative effects of a tax increase is inaccurate or exaggerated	89 (1.1)
There is no proof of a link between the measure and smuggling or the dark market	59 (0.7)
There is no proof of a link between the measure and increased risk for tobacconists	16 (0.2)
The general population is supportive of the measure	14 (0.2)

**Table 4 ijerph-19-15152-t004:** Alternative proposals or prerequisites to tobacco tax increase.

Category of Arguments (m = 8015 Arguments Spread across 3501 Articles)	Occurrences, *n* (%)
Alternatives or prerequisites	284 (3.5)
(H) The taxation strategy alone is not effective to reduce smoking—it needs to work on the back of other measures	141 (1.8)
Prices should be harmonized within the European Union	85 (1.1)
We need to be tougher in the fight against smuggling and the dark market	28 (0.3)
We should be developing prevention measures for youth instead of increasing taxes	22 (0.3)
We should be advocating lower-risk products as a way to quit smoking instead of increasing taxes	6 (0.07)
(I) A tax increase will benefit all if moderate (defends an increase limited at 5%)	143 (1.8)

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
