# Peer review of "Tobacco Tax Increases: A Discourse Analysis of the French Print and Web News Media from 2000 to 2020"

_ijerph, 2022, doi:10.3390/ijerph192215152_

Round 1

Reviewer 1 Report

Thank you for the opportunity to review this work. 

In my opinion, this is a very important and well-prepared manuscript. I congratulate you for undertaking this meticulous analysis and am full of appreciation for the clear presentation of the results. Methodologically, this is a very well-designed study that I read with considerable admiration. When reading the "Identified Arguments" section, I had to reread because the number of arguments and their additional breakdown made me have to go back and analyze the results presented. For me, however, this is an asset, not a bug. The authors offered a complex argumentation, but this is normal when analyzing such extensive material.

In the discussion section, the authors pointed out in a very understandable way (indicating the cause-effect sequence) in the analyzed publications the existence of imbalances in the discourse on increasing the tobacco tax, which delay the introduction of increases affecting the expected reduction of the phenomenon.

The purpose of this article was to examine the different types of arguments used around tobacco tax increases disseminated by various opponents and supporters through the French general press. In my opinion, the goal has been met.

The recommendations in the concluding paragraph of the "Conclusions" section should be taken into account, but I realize how challenging it is to control what editors and journalists undertake to cover in their journals.  Won't this result in a battle over violations of freedom of speech? These are just my thoughts, however. I hope this publication will get someone's attention, however. In my opinion this is very important topic.

Author Response

AUTHORS REPONSE

Important and difficult comment! We tried to take into consideration your comments by adding a line and references on tobacco control advocates’ responsibility to provide evidence based arguments through media, in key legislative periods, so decision makers can take part to the debate with all information at hand. Our results encourage them to be more proactive and professional in their media advocacy strategies. The point is more to invite NGO’s to rethink their strategies more than denying other stakeholders right or legitimacy to enter the debate via newspaper.

In the conclusion section

[…] Media is a key tool for tobacco control advocacy, both to inform legislators and the wider public and to support the adoption of policies [39,40]. Our research reveals that French NGOs should develop more proactive and professionalized press relations: […] (p13)

References

MacKenzie, R.; Chapman, S. Generating News Media Interest in Tobacco Control; Challenges in an Advanced Policy Environment. Health Promot J Aust 2012, 23, 92–96, doi:10.1071/HE12092.

Lane, C.H.; Carter, M.I. The Role of Evidence-Based Media Advocacy in the Promotion of Tobacco Control Policies. Salud Publica Mex 2012, 54, 281–288.

Reviewer 2 Report

 This manuscript is well written. The methodology used is adequate and the results are well presented and discussed.  Specific comments follow:

1.Regarding the international interest of the study, its interest should be better highlighted. Currently, negotiations and discussions upon the review of the EU TOBACCO TAXES DIRECTIVE are being developed. This should be highlighted in the introduction. 

2. I suggest that authors cite the paper indicated below since it is highly important to frame the discussion upon tobacco taxes in the European Union. Promoting convergence and closing gaps using affordability-based minimum taxes: an illustration using the European Union Tobacco Tax Directive. Branston JR, López-Nicolás Á. Tob Control. 2022 Feb 11:tobaccocontrol-2021-056960. doi: 10.1136/tobaccocontrol-2021-056960. Epub ahead of print. PMID: 35149599.

Author Response

Comment 1

Regarding the international interest of the study, its interest should be better highlighted. Currently, negotiations and discussions upon the review of the EU TOBACCO TAXES DIRECTIVE are being developed. This should be highlighted in the introduction. 

AUTHOR RESPONSE

Thank you for your comment. We added information on the European Directive in the introduction. We highlighted the conclusions of the 2020 evaluation of the current Directive on tobacco taxation that are in line with the scientific evidences showing that a weak taxation policy has no impact on tobacco use.

Introduction section

[…] The research reported here is conducted in Europe, where the ‘Tobacco Tax’ Directive 2011/64/EU (TTD) defines a minimum excise duty rates (between 7.5% and 76.5% of the total tax burden) and an ad valorem tariff expressed as a percentage of the maximum pack price[11]. The TTD was recently evaluated and found to perform ineffectively on  deterring tobacco use, and so the European Commission is currently considering a more ambitious harmonized taxation policy [12]. As the TTD is not fully harmonized, some Member States have used specific tobacco policy taxation embedded in the Directive.

References

Excise Duties on Tobacco Available online: https://taxation-customs.ec.europa.eu/taxation-1/excise-duties/excise-duties-tobacco_en (accessed on 10 November 2022).

Revision of Excise Rules for Tobacco Available online: https://taxation-customs.ec.europa.eu/revision-excise-rules-tobacco_en (accessed on 10 November 2022).

Comment 2

I suggest that authors cite the paper indicated below since it is highly important to frame the discussion upon tobacco taxes in the European Union. Promoting convergence and closing gaps using affordability-based minimum taxes: an illustration using the European Union Tobacco Tax Directive. Branston JR, López-Nicolás Á. Tob Control. 2022 Feb 11:tobaccocontrol-2021-056960. doi: 10.1136/tobaccocontrol-2021-056960. Epub ahead of print. PMID: 35149599.

AUTHOR RESPONSE

We thank you for your comment. We included it in the discussion section (see below):

In the discussion section

[…] Another limitation is that our research is France-centered. France is part of the wider the European Union where the TTD is currently being revised. Some researchers and public health figures propose using the TTD revision to converge prices and adopt an affordability-based minimum tax [37]. In this context, it would be instructive to monitor the main European media channels to capture the arguments developed by the tobacco industry to counter this proposal. […] (p13)

References

Branston, J.R.; López-Nicolás, Á. Promoting Convergence and Closing Gaps Using Affordability-Based Minimum Taxes: An Illustration Using the European Union Tobacco Tax Directive. Tob Control 2022, tobaccocontrol-2021-056960, doi:10.1136/tobaccocontrol-2021-056960.